# Orthogonal Subspace Learning for Language Model Continual Learning

**Xiao Wang**[★][*], **Tianze Chen**[★][*], **Qiming Ge**[★], **Han Xia**[★],
**Rong Bao**[★], **Rui Zheng**[★], **Qi Zhang**[★][†], **Tao Gui**[♦][†] **Xuanjing Huang**[★][♣]

[★] School of Computer Science, Fudan University, Shanghai, China
[♦] Institute of Modern Languages and Linguistics, Fudan University, Shanghai, China
[♣] International Human Phenome Institutes (Shanghai)
{xiao_wang20,qz,tgui}@fudan.edu.cn

## Abstract

Benefiting from massive corpora and advanced hardware, large language models (LLMs) exhibit remarkable capabilities in language understanding and generation. However, their performance degrades in scenarios where multiple tasks are encountered sequentially, also known as catastrophic forgetting. In this paper, we propose orthogonal low-rank adaptation (O-LoRA), a simple and efficient approach for continual learning in language models, effectively mitigating catastrophic forgetting while learning new tasks. Specifically, O-LoRA learns tasks in different (low-rank) vector subspaces that are kept orthogonal to each other in order to minimize interference. Our method induces only marginal additional parameter costs and requires no user data storage for replay. Experimental results on continual learning benchmarks show that our method outperforms state-of-the-art methods. Furthermore, compared to previous approaches, our method excels in preserving the generalization ability of LLMs on unseen tasks.

## 1 Introduction

Learning tasks sequentially is crucial for developing real-world NLP models (Wang et al., 2023b; Xi et al., 2023), as it enables continuous evolution when encountering new tasks or knowledge. Although pre-trained models (Devlin et al., 2019; Brown et al., 2020; Raffel et al., 2020; OpenAI, 2023) have achieved tremendous success on static tasks (Wang et al., 2022a, 2023c), learning multiple tasks sequentially, commonly referred to as continual learning, remains challenging (Wu et al., 2022; Luo et al., 2023). As a model learns new tasks, it tends to forget or lose the knowledge it had acquired for earlier tasks, leading to a phenomenon known as catastrophic forgetting (McCloskey and Cohen, 1989).

---

[*] Equal contribution
[†] Corresponding Author

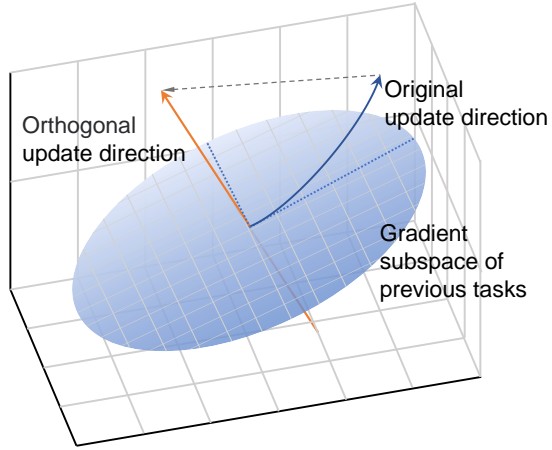

Figure 1: Illustration highlighting the intuition of our approach. O-LoRA mitigates catastrophic forgetting of past task knowledge by constraining the gradient updates of the current task to be orthogonal to the gradient subspace of the past tasks.

Existing continual learning works (Ke and Liu, 2022; Wang et al., 2023a) can be mainly categorized into rehearsal-based, regularization-based, and architecture-based approaches. Rehearsal-based approaches (Lopez-Paz and Ranzato, 2017; de Masson D'Autume et al., 2019) allow access to a memory buffer with examples from prior tasks and train the model jointly with the current task. Unfortunately, storing and replaying data from previous tasks may raise privacy concerns, especially when sensitive or personally identifiable information is involved. Regularization-based approaches (Kirkpatrick et al., 2017; Li and Hoiem, 2017; Smith et al., 2023) introduce additional terms in the loss function to penalize changes in important weights, aiming to protect earlier learned tasks. They often struggle to handle long task sequences. Architecture-based approaches (Wang et al., 2023e; Razdaibiedina et al., 2023) dynamically expand the model capacity or isolate existing model weights to reduce interference.

However, such approaches essentially learn different expert models for different tasks, limiting their generalization to unseen tasks.

Existing methods typically update all tasks within a shared vector space, directly affecting the model's hidden layer outputs. Recent studies (Farajtabar et al., 2020; Saha et al., 2021) have highlighted a promising approach to address this issue. By taking gradient steps in the orthogonal direction to the gradient subspaces associated with past tasks, we can effectively mitigate catastrophic forgetting as it prevents interference with the past task loss functions. However, previous approaches either require storing historical data (Chaudhry et al., 2019), which raises data privacy concerns, or historical data gradients (Farajtabar et al., 2020), which becomes impractical for large-scale models.

In this work, we propose orthogonal low-rank adaptation (O-LoRA) [1], a simple and efficient approach for continual learning in language models. Our key insight is rooted in the nature of LoRA: large pre-trained models primarily fine-tune within a specific low-rank subspace. With this premise, we hypothesize that the gradient subspaces from previous tasks can be effectively captured by the LoRA parameters. In the context of continual learning, we incrementally learn new tasks in an orthogonal subspace while fixing the LoRA parameters learned from past tasks. Figure 1 provides a visual representation of how O-LoRA minimizes catastrophic forgetting.

Our method offers three advantages: (1) **Data privacy-friendliness**: We require no storage of user data for replay, addressing concerns associated with privacy. (2) **Model parameter-friendliness**: By introducing only marginal cost of additional parameters, our approach enables the learning of new tasks without compromising the performance of previous tasks. (3) **Generalization-friendliness**: Our method does not rely on task IDs during testing, making it compatible with instruction tuning paradigm (Wang et al., 2022b), thus preserving LLMs' generalization ability on unseen tasks.

Our main contributions are summarized as follows:

- We introduce O-LoRA, a simple and efficient approach for continual learning in language

models, incrementally learning new tasks in orthogonal subspaces.

- Our method significantly outperforms prior SOTA methods on standard continual learning benchmarks.

- Experimental results show that our method preserves the generalization ability of large language models on unseen tasks, which was lacking in previous approaches.

## 2 Background

### 2.1 Continual Learning Setup

Continual learning (Ke and Liu, 2022; Wang et al., 2023b) focuses on developing learning algorithms to accumulate knowledge on non-stationary data. In supervised continual learning, a sequence of tasks $\{\mathcal{D}_1, \ldots, \mathcal{D}_T\}$ arrive in a streaming fashion. Each task $\mathcal{D}_t = \left\{\left(\boldsymbol{x}_i^t, y_i^t\right)\right\}_{i=1}^{n_t}$ contains a separate target dataset, where $\boldsymbol{x}_i^t \in \mathcal{X}_t$, $\boldsymbol{y}_i^t \in \mathcal{Y}_t$. A single model needs to adapt to them sequentially, with only access to $\mathcal{D}_t$ at the t-th task. In general, given a prediction model $h_\Theta$ parameterized by $\Theta$, continual learning seeks to optimize for the following objective across all tasks:

$$\max_\Theta \sum_{k=1}^{T} \sum_{x,y \in \mathcal{D}_k} \log p_\Theta(y \mid x) \qquad (1)$$

In this study, we tackle a more challenging setting. During the training phase, the model is prohibited from accessing any historical data. In the testing phase, the model predicts a sample's label without knowing which task it belongs to.

### 2.2 LoRA

When pre-trained models (PTMs) adapt to specific tasks, Hu et al. (2021) has demonstrated that weight updates in PTMs exhibit a low "intrinsic dimension." For a pre-trained weight matrix $W_{init} \in R^{d \times k}$, LoRA constrains its update by representing it with a low-rank decomposition $W_{init} + \Delta W = W_{init} + AB$, where $A \in R^{d \times r}$, $B \in R^{r \times k}$, and the rank $r \ll min(d, k)$. $W_{init}$ remains fixed during training and does not receive gradient updates, while $A$ and $B$ contain trainable parameters. To illustrate the modified forward pass of LoRA, consider the operation $h = W_{init}x$. With LoRA, the modified forward pass becomes:

$$h = W_{init}x + \Delta W x = W_{init}x + ABx \qquad (2)$$

---

[1]The dataset, code can be found at https://github.com/cmnfriend/O-LoRA

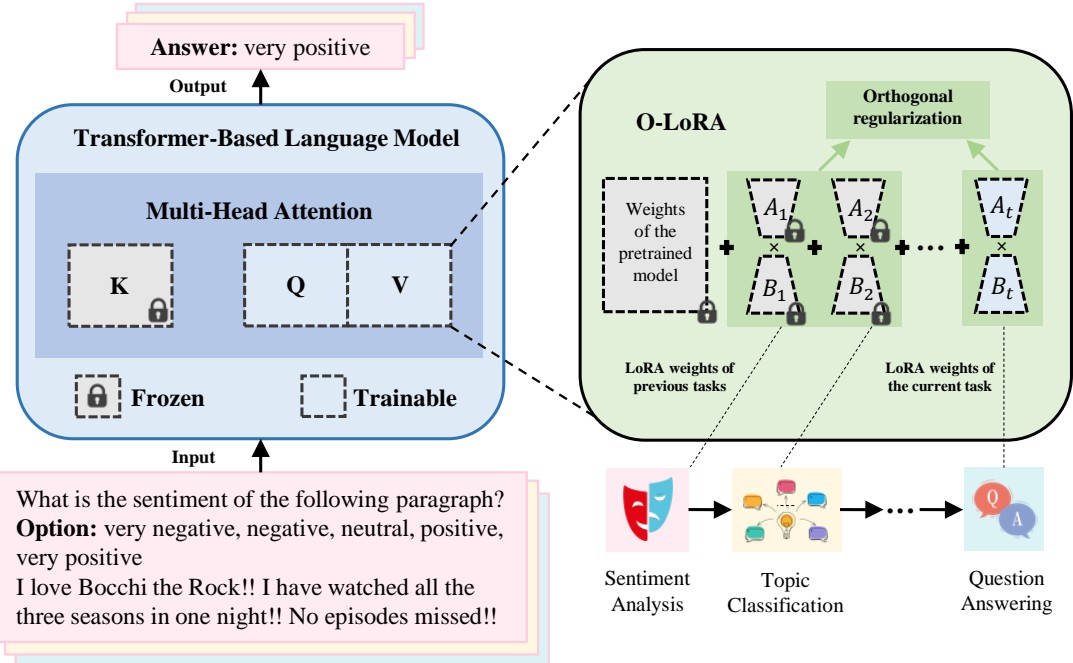

Figure 2: The framework of O-LoRA for language model continual learning. First, allowing the integration of human expertise and enhancing generalization by instruction tuning. Next, approximate gradient subspaces of each task respectively uning LoRA. For each sequentially incoming task, we incrementally learn a new LoRA while enforcing orthogonality between the current task's LoRA and the past ones.

## 3   Orthogonal Low-rank Adaptation

In this section, we introduce O-LoRA, illustrated in Figure 2. First, we adopt instruction tuning as our training paradigm. Then, we incrementally learn new tasks within an orthogonal subspace while keeping the LoRA parameters fixed for past tasks. Lastly, we conduct a comparative analysis of our method compared to existing approaches.

### 3.1   Instruction Schema

Instruction-following capability is essential to LLMs as an interface between humans and AI models (Wang et al., 2022b; Ouyang et al., 2022; Wang et al., 2023d).   We choose instruction tuning as our training paradigm for two reasons: 1) Incorporating human expertise: The models can leverage prior knowledge and benefit from human expertise by providing explicit instructions, leading to more efficient learning. 2) Enhanced generalization: The explicit guidance helps models capture the underlying principles, enabling better generalization to unseen situations.

All task instructions follow the same uniform schema, which is composed of 1) **Task Definition** provides a detailed guide on how an input text (e.g., a sentence or a document) is expected to be mapped

to an output text. 2) **Options** are the output label constraints for a task, which represent the set of possible outputs that can be generated by the model for a given input. 3) **Text** is the input sentence of a task instance.  This sequence is then fed into the pre-trained language model along with the task instruction and options. 4) **Answer** is the expected output of the given sample.

### 3.2   Continual Learning in Orthogonal Subspaces

Previous methods exhibit a common feature: all tasks undergo updates within a shared vector space, directly impacting the hidden layer outputs of the model. Catastrophic forgetting happens in neural networks when the gradient updates with respect to a new task are applied to the model without considering previous tasks.

Farajtabar et al. (2020) propose the Orthogonal Gradient Descent (OGD) method for mitigating this problem, which constrains the parameters to move within the orthogonal space to the gradients of previous tasks. With limited access to previous task data, OGD approximates the current gradient of previous data with the gradient in the previous convergence parameters.  However, OGD needs to store gradients of all previous data.  This can

be especially intractable for large-scale language models with billions of parameters (Raffel et al., 2020; Brown et al., 2020), which have become a standard in the NLP field.

*Is it possible to approximate the gradient direction of previous tasks without storing historical gradients?* In this study, we leverage the low-rank subspace of LoRA (Hu et al., 2021) as a proxy for the gradient subspace of past tasks. Our fundamental insight is rooted in the nature of LoRA: large pre-trained models primarily fine-tune within a specific low-rank subspace. This characteristic behavior suggests that the LoRA parameters are not mere numerical adjustments but encapsulate crucial model update directions. Therefore, we hypothesize that the gradient subspaces of previous tasks are succinctly represented by the LoRA parameters. By learning within a subspace orthogonal to the LoRA subspace associated with previous tasks, we can prevent interference with past task loss functions, thus mitigating catastrophic forgetting.

We propose O-LoRA, which incrementally learns new tasks in a direction orthogonal to the LoRA subspace of past tasks while fixing the previous parameters. For each task, we introduce a set of LoRA parameters denoted as $\{A_t, B_t\}$, where $A \in \mathbb{R}^{d \times r}$, $B \in \mathbb{R}^{r \times k}$, and the rank $r \ll \min(d, k)$. We approximate the parameter update subspace $\mathcal{U}_t$ for the $t$-th task as the subspace spanned by the column vectors of $A_t$:

$$A_t = [a_t^1, a_t^2, ..., a_t^r] \qquad (3)$$

$$\mathcal{U}_t = \text{span}\{a_t^1, a_t^2, ..., a_t^r\} \qquad (4)$$

Let $B_t = [b_t^1, b_t^2, ..., b_t^r]$, where $b_t^i \in B_t$ represents the linear weighting coefficients of the column vectors in $A_t$.

To ensure the orthogonality between the subspace $\mathcal{U}$ and the subspace $\mathcal{W}$, we need to satisfy:

$$<u, w> = 0, \forall u \in \mathcal{U}, w \in \mathcal{W}. \qquad (5)$$

Therefore, achieving orthogonality between the LoRA subspaces of task $i$ ($\mathcal{U}^i$) and task $t$ ($\mathcal{U}^t$) can be expressed as:

$$O_{i,t} = A_i^T A_t = 0. \qquad (6)$$

Finally, our training objective is defined as:

$$\sum_{x,y \in \mathcal{D}_t} \log p_\Theta(y \mid x) + \lambda_1 \sum_{i=1}^{t-1} L_{orth}(A_i, A_t) \quad (7)$$

| | RF | PE | TIF | UT |
|---|---|---|---|---|
| EWC (Kirkpatrick et al., 2017) | ✓ | | ✓ | |
| A-GEM (Chaudhry et al., 2018) | | | ✓ | |
| MBPA++ (de Masson D'Autume et al., 2019) | | | ✓ | |
| IDBR (Huang et al., 2021) | | | ✓ | |
| L2P (Wang et al., 2022c) | ✓ | ✓ | ✓ | |
| LwF (Li and Hoiem, 2017) | ✓ | | | |
| OGD (Farajtabar et al., 2020) | ✓ | | ✓ | |
| LFPT5 (Qin and Joty, 2021) | | ✓ | ✓ | |
| EIP (Wang et al., 2023e) | ✓ | ✓ | ✓ | |
| PP (Razdaibiedina et al., 2023) | ✓ | ✓ | | |
| **O-LoRA** | ✓ | ✓ | ✓ | ✓ |

Table 1: The comparison between O-LoRA and other continual learning methods. Specifically, **RF** indicates whether the method is rehearsal-free. **PE** indicates whether the method is parameter efficient. **TIF** indicates whether task-id is available during inference. **UT** indicates whether the method can be applied to solve unseen tasks.

$$L_{orth}(A_i, A_t) = \sum_{j,k} \|O_{i,t}[j, k]\|^2 \qquad (8)$$

where $O_{i,t}[j, k]$ denotes the element at the $j$-th row and $k$-th column of $O_{i,t}$, and $\lambda_1$ is the weights of the orthogonality loss. During the training process, to mitigate forgetting of past knowledge, we fix the previous LoRA parameters $\{A_i, B_i | i < t\}$. Following Hu et al. (2021), we only apply LoRA to the attention weights of queries ($W_q$) and values ($W_v$).

While the number of LoRA parameters grows with the number of tasks during training, we can merge the updates corresponding to the LoRA parameters into the initial parameters to avoid GPU memory inflation.

$$W_{init} := W_{init} + \sum_{i=1}^{t} A_i B_i. \qquad (9)$$

### 3.3 Comparisons Between O-LoRA and Other Methods

In this section, we compare O-LoRA with other existing continual learning methods across several dimensions: rehearsal-free, parameter efficiency, availability of task-id during inference, and applicability to unseen tasks. As shown in Table 1, O-LoRA demonstrates three distinct advantages: data privacy-friendliness, model parameter-friendliness, and generalization-friendliness.

**Data privacy-friendliness**. Rehearsal-based methods (de Masson D'Autume et al., 2019; Huang et al., 2021), which rely on storing past task data in a buffer and replaying it during training,

are not suitable for scenarios with data privacy concerns. Additionally, as the number of training tasks increases, the cost of training new tasks using replay-based methods also grows. In contrast, our method does not require storing historical data, alleviating concerns regarding data privacy. Moreover, since we only modify the training loss, there is no additional training cost incurred.

**Model parameter-friendliness**. Many previous methods (Kirkpatrick et al., 2017; Farajtabar et al., 2020) train the entire model parameters for each task, while our method only introduces marginal additional parameters for each task. O-LoRA has lower requirements in terms of computational resources and GPU memory during training. Additionally, because the training of LoRA freezes the pre-trained model parameters, it is less prone to forgetting the knowledge acquired during pre-training.

**Generalization-friendliness**. Traditional methods (Kirkpatrick et al., 2017; Chaudhry et al., 2018; Wang et al., 2022c), primarily designed for classification tasks, often fall short in generalizing to unseen tasks due to their narrow task-specific focus. In contrast, O-LoRA employs instruction tuning (Wang et al., 2022b) as its training paradigm. By incorporating explicit instructions or demonstrations, the model can capture the underlying principles or constraints of a task. This explicit guidance helps the model generalize beyond the specific examples in the training data, enabling it to handle unseen situations more effectively. The integration of human expertise through instruction tuning enhances the generalization capabilities of O-LoRA.

## 4 Experiments

### 4.1 Experimental Setup

#### 4.1.1 Datasets

**Standard CL Benchmark** We evaluate our approach using the CL benchmark for language models, which consists of five text classification datasets introduced by Zhang et al. (2015): AG News, Amazon reviews, Yelp reviews, DBpedia and Yahoo Answers. We adopt the CL setup for the T5 model, following LFPT5 (Qin and Joty, 2021), and explore three different orders of the benchmark. Appendix A.2 provides the task details, and the sequences of tasks used in our experiments are provided in Appendix A.3.

**Large number of tasks** Our method's performance on longer task sequences, posing a greater challenge, is evaluated through experiments on a continual learning benchmark of 15 datasets (Razdaibiedina et al., 2023). This includes five tasks from CL benchmark, four from GLUE benchmark (MNLI, QQP, RTE, SST2) (Wang et al., 2018), five from SuperGLUE benchmark (WiC, CB, COPA, MultiRC, BoolQ) (Wang et al., 2019), and the IMDB movie reviews dataset (Maas et al., 2011). Following Razdaibiedina et al. (2023), we select 1000 random samples for training each task and hold out 500 samples per class for validation.

**Unseen tasks Generation** To assess the impact of our approach on LLMs' generalization ability, we initially train an LLM on the Alpaca dataset (Taori et al., 2023), an open-source multitask instruction tuning dataset. We then use the pre-trained LLM for sequential training on the standard CL benchmark (Zhang et al., 2015). Our zero-shot benchmark, MMLU (Hendrycks et al., 2020), covers 57 subjects across various domains such as STEM, humanities, and social sciences, assessing world knowledge and problem-solving abilities across various difficulty levels.

#### 4.1.2 Metrics

Let $a_{i,j}$ be the testing accuracy on the i-th task after training on j-th task, the metrics for evaluating is **Average Accuracy (AA)**, the average accuracy of all tasks after training on the last task, $\frac{1}{T} \sum_{i=1}^{T} a_{i,T}$

#### 4.1.3 Baselines

We evaluate O-LoRA against 10 baseline methods. Importantly, among these baselines, only prompt-based methods are exceptions; all others utilize the LoRA framework. This uniformity in the foundation ensures consistent parameter settings between O-LoRA and its comparatives, guaranteeing a fair comparison.

- **SeqFT** (de Masson D'Autume et al., 2019): train all model parameters on a sequence of tasks (without adding any regularization or replaying samples from the previous tasks).

- **SeqLoRA**: fixed-size LoRA parameters are trained on a sequence of tasks (without adding any regularization or replaying samples from the previous tasks).

- **IncLoRA**: incremental learning of new LoRA parameters on a sequential series of tasks

| | Standard CL Benchmark | | | | Large Number of Tasks | | | |
|---|---|---|---|---|---|---|---|---|
| | **Order-1** | **Order-2** | **Order-3** | **avg** | **Order-4** | **Order-5** | **Order-6** | **avg** |
| SeqFT | 18.9 | 24.9 | 41.7 | 28.5 | 7.4 | 7.4 | 7.5 | 7.4 |
| SeqLoRA | 44.6 | 32.7 | 53.7 | 43.7 | 2.3 | 0.6 | 1.9 | 1.6 |
| IncLoRA | 66 | 64.9 | 68.3 | 66.4 | 63.3 | 58.5 | 61.7 | 61.2 |
| Replay | 55.2 | 56.9 | 61.3 | 57.8 | 55 | 54.6 | 53.1 | 54.2 |
| EWC | 48.7 | 47.7 | 54.5 | 50.3 | 45.3 | 44.5 | 45.6 | 45.1 |
| LwF | 54.4 | 53.1 | 49.6 | 52.3 | 50.1 | 43.1 | 47.4 | 46.9 |
| L2P | 60.3 | 61.7 | 61.1 | 60.7 | 57.5 | 53.8 | 56.9 | 56.1 |
| LFPT5 | 67.6 | 72.6 | **77.9** | 72.7 | 70.4 | **68.2** | 69.1 | 69.2 |
| **O-LoRA** | **75.4** | **75.7** | 76.3 | **75.8** | **72.3** | 64.8 | **71.6** | **69.6** |
| ProgPrompt | 75.2 | 75 | 75.1 | 75.1 | 78.0 | 77.7 | 77.9 | 77.9 |
| PerTaskFT | 70.0 | 70.0 | 70.0 | 70.0 | 78.1 | 78.1 | 78.1 | 78.1 |
| MTL | 80.0 | 80.0 | 80.0 | 80.0 | 76.5 | 76.5 | 76.5 | 76.5 |

Table 2: Summary of the results on two standard CL benchmarks with T5-large model. Averaged accuracy after training on the last task is reported. All results are averaged over 3 runs.

(without adding any regularization or replaying samples from the previous tasks).

- **Replay**: finetune the whole model with a memory buffer, and replay samples from old tasks when learning new tasks to avoid forgetting.

- **EWC** (Kirkpatrick et al., 2017): finetune the whole model with a regularization loss that prevents updating parameters that could interfere with previously learned tasks.

- **LwF** (Li and Hoiem, 2017): constrains the shared representation layer to be similar to its original state before learning the new task.

- **L2P** (Wang et al., 2022c): uses the input to dynamically select and update prompts from the prompt pool in an instance-wise fashion.

- **LFPT5** (Qin and Joty, 2021): continuously train a soft prompt that simultaneously learns to solve the tasks and generate training samples, which are subsequently used in experience replay.

- **ProgPrompt** (Razdaibiedina et al., 2023): adopts a task-specific soft prompt for each distinct task, sequentially appending it to prior learned prompts. In essence, it trains individual models per task, leveraging the task ID to select the appropriate model during inference.

- **PerTaskFT**: train a separate model for each task.

- **MTL**: train a model on all tasks as multi-task learning. This method is the upper bound of continual learning.

#### 4.1.4 Implementation Details

O-LoRA is a model-agnostic CL method that can be used with any transformer-based model. In our experiments, we use two language models adopted by the previous lines of works in CL for NLP: encoder-decoder T5 model (Raffel et al., 2020) and decoder-only LLaMA model (Touvron et al., 2023). To compare O-LoRA to the recent CL approaches (Wang et al., 2022c; Qin and Joty, 2021), we use the pre-trained T5-large model. To validate the impact of our approach on the generalization ability of LLMs for unseen tasks, we use pre-trained LLaMA-7B model. All experimental results are reported as the average of 3 runs. For more detailed settings, refer to the Appendix A.1.

### 4.2 Main Results

Table 2 presents a performance comparison of O-LoRA and baseline continual learning methods on two CL benchmarks. Following LFPT5, we report the results of three independent runs with different task orders on the CL benchmark.

**Results on Standard Continual Learning Benchmarks** On all task orders of the standard CL benchmark, O-LoRA consistently outperforms previous methods by a significant margin. Overall, O-LoRA achieves a performance improvement of over 24% compared to LFPT5, the previous state-

|              | MMLU | CL   |
|--------------|------|------|
| LLaMA-7B     | 34.4 | /    |
| Alpaca-LoRA  | 37.5 | /    |
| Alpaca-LoRA-CL | 23.3 | 46.7 |
| Alpaca-inc-LoRA-CL | 28.6 | 33.1 |
| Alpaca-OLoRA-CL | 33.6 | 76.8 |

Table 3: Performance comparison of different continual learning methods applied to the Alpaca-LoRA-LLaMA model. These methods are evaluated on MMLU(zero-shot) and CL benchmmark(order 1).

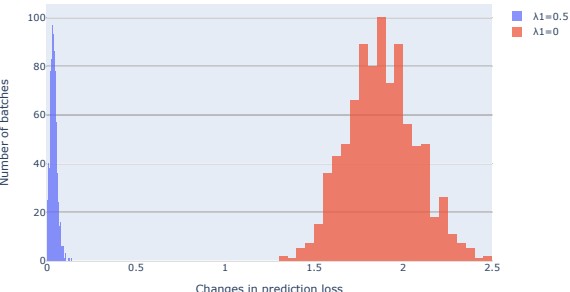

Figure 3: Histogram of prediction loss changes after training on a new task. The O-LoRA constraint ($\lambda_1 = 0.5$) helps reduce the changes in comparison to when it is not present ($\lambda_1 = 0$).

of-the-art method. Our approach demonstrates comparable performance to multi-task learning and significantly outperforms PerTaskFT, indicating that our method not only effectively avoids catastrophic forgetting but also leverages knowledge from past tasks for efficient learning of new tasks.

**Performance with Large Number of Tasks** On a more challenging benchmark with a large number of tasks, O-LoRA outperforms the state-of-the-art method, LFPT5, in terms of the average performance across three orders of tasks. While ProgPrompt performs better than our method in handling long sequence tasks, its inherent constraints cannot be overlooked. ProgPrompt is strictly tied to tasks it's trained on and leans heavily on task IDs during inference, limiting its generalization ability and making it less adaptive for LLMs. It is worth noting that almost all existing continual learning methods perform significantly lower than PerTaskFT and MTL, indicating that continual learning for a large number of tasks remains a challenging problem.

**Impact on the Generalization Ability of LLMs** We investigate the impact of O-LoRA on the generalization of Large Language Models through continual learning experiments. We start with a fine-tuned LLaMA-7B language model on the Alpaca dataset, then test models with and without the O-LoRA constraint on the MMLU benchmark. With MMLU being a four-classification problem, a 25% accuracy equates to random guessing. According to Table 3, models without O-LoRA (Alpaca-LoRA-CL, Alpaca-LoRA-inc-CL) achieve accuracies of 23.3% and 28.6% respectively, comparable to random guesses. In contrast, models with O-LoRA average at 33.6% accuracy, demonstrate the effectiveness of O-LoRA in maintaining generalization for unseen tasks.

### 4.3 Discussions

**Does O-LoRA preserve the loss of previous tasks while training new tasks?** We assessed the efficiency of the low-rank subspace of LoRA in approximating the gradient subspace of previous tasks. In our evaluation, we applied an orthogonality constraint with a weight of 0.5 ($\lambda_1 = 0.5$). In comparison, without the constraint ($\lambda_1 = 0$), new LoRA parameters are added for new tasks with historical LoRA, and model parameters are kept fixed. As Figure 3 shows, the O-LoRA constraint helps keep the loss of previous samples low, proving that the O-LoRA constraint effectively counteracts catastrophic forgetting.

**How does O-LoRA influence the output of each layer in the model?** We examine the variation in hidden states for past task samples in models trained with and without O-LoRA constraints, using the T5-base model. Figure 4 demonstrates that the O-LoRA constraint minimizes the variations, hence reducing the forgetting of internal knowledge. We found that lower layers encode more generic semantic knowledge that can be shared across tasks. Conversely, higher layers encode task-specific semantic knowledge and change a lot during new task learning. The decoder can capture relevant information from these rich semantic representations, proving the minimal impact of our method on past tasks.

**How do different PLMs influence performances?** We evaluate the performance of models across varying parameter sizes (T5-base, T5-large, T5-XL) and distinct architectures (T5, LLaMA) using the standard continual learning benchmark. Our findings are as follows: 1) In the T5 series, O-LoRA's average accuracy improves as the parameter size increases. 2) Larger network

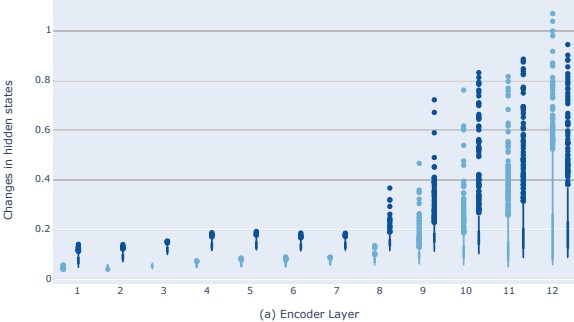

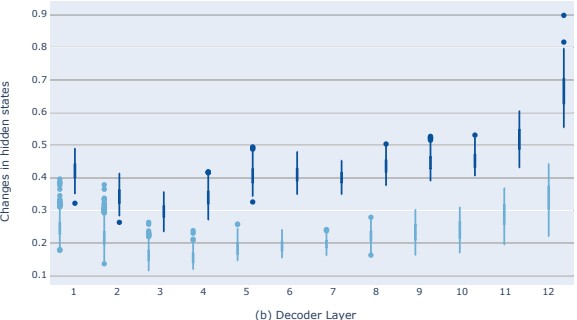

Figure 4: Variation in hidden states across different layers of the T5-base model with and without O-LoRA constraints. (a) illustrates the changes in the hidden states of each layer in the T5 encoder after training on a new task. (b) demonstrates the changes in the hidden states of each layer in the T5 decoder. Light blue color indicates the utilization of orthogonality loss, while dark blue represents the absence of orthogonal loss in the training objective.

sizes appear to counteract catastrophic forgetting, approaching the proficiency levels of multitask learning. 3) Notably, even with a greater parameter count in the LLaMA-7B model, the T5-3B model registered a higher average accuracy. This implies that encoder-decoder architectures might be more resistant to forgetting.

**What is the Optimal Rank r for O-LoRA?** To investigate the influence of the rank parameter (r) on the performance of O-LoRA, we conduct experiments using T5-Base on a standard CL benchmark. Table 5 presents the results of varying r values. Increasing the rank r improves the average accuracy of the model to a certain extent. However, we observe that there is not a significant difference in performance between r=2 and r=16, indicating that the gradient space of the model has a relatively low intrinsic dimensionality.

| Model | Order | | | | |
| | 1 | 2 | 3 | avg | MTL |
|---|---|---|---|---|---|
| T5-base | 73.9 | 75.8 | 74.5 | 74.7 | 78.5 |
| T5-large | 75.4 | 75.7 | 76.3 | 75.8 | 80.0 |
| T5-xl | 78.9 | 79.0 | 77.9 | 78.6 | 79.9 |
| LLaMA-7B | 76.8 | 75.7 | 75.7 | 76.1 | 77.1 |

Table 4: Comparison of different PLMs' performances across three orders in a standard continual learning benchmark. Results also include average accuracy ("avg") and multitask learning performance ("MTL").

| r-dim | Order | | | |
| | 1 | 2 | 3 | avg |
|---|---|---|---|---|
| 2 | 74.2 | 71.1 | 73.6 | 73.0 |
| 4 | 73.0 | 72.7 | 74.1 | 73.3 |
| 8 | 75.6 | 71.7 | 71.9 | 73.1 |
| 16 | 74.5 | 73.4 | 74.8 | 74.2 |
| **std** | 0.92 | 0.89 | 1.07 | 0.47 |

Table 5: Comparisons of different rank r of LoRA. This experiment is conducted based on T5-Base on the standard continual learning benchmark.

## 5 Related Work

### 5.1 Continual Learning

Continual learning (Ke and Liu, 2022; Wang et al., 2023a) aims to develop learning algorithms that can accumulate knowledge on non-stationary data. Existing works can be broadly categorized into rehearsal-based, regularization-based, and architecture-based approaches. For an in-depth discussion on continual learning in the era of large language models, readers may refer to (Wang et al., 2023b).

**Rehearsal-based approaches** (Lopez-Paz and Ranzato, 2017; de Masson D'Autume et al., 2019; Han et al., 2020; Bai et al., 2022) leverage a memory buffer that stores examples from previous tasks, training the model jointly with the current task. Experience replay (ER) (Rolnick et al., 2019) is a common strategy employed in rehearsal-based approaches and serves as a strong baseline. However, the storage and replay of data from previous tasks raise privacy concerns, particularly when dealing with sensitive information.

**Regularization-based approaches** (Kirkpatrick et al., 2017; Li and Hoiem, 2017; Farajtabar et al., 2020; Smith et al., 2023) incorporate additional

terms into the loss function to penalize changes in crucial weights. For instance, Orthogonal Gradient Descent (OGD) (Farajtabar et al., 2020) constrains the parameters to move within the orthogonal space defined by the gradients of previous tasks. However, OGD requires storing gradients of all historical data, which becomes infeasible for large language models. Another work introduces C-LoRA (Smith et al., 2023) for continual learning of text-conditioned images, which regularizes the similarity of new LoRA parameters with historical versions, limiting their learning plasticity to new tasks.

**Architecture-based approaches** (Wang et al., 2023e; Razdaibiedina et al., 2023) focus on dynamically expanding model capacity or isolating existing model weights to mitigate interference between new and old tasks. Progressive Prompts (Razdaibiedina et al., 2023) learns separate prompts for each incoming task and sequentially concatenates them with previously learned prompts. However, such approaches essentially train distinct expert models for different tasks, which restricts their generalization ability to unseen tasks.

In contrast to existing methods, our approach offers unique advantages in terms of data privacy, model parameter efficiency, and generalization capability, as discussed in the previous sections.

### 5.2 Parameter Efficient Tuning

Parameter Efficient Tuning (PET) (He et al., 2021) has emerged as a significant research direction aimed at optimizing model performance while minimizing computational resources and annotation efforts. Various approaches have been proposed to achieve parameter efficiency in tuning, including adapters (Houlsby et al., 2019), prompt learning (Lester et al., 2021), LoRA (Hu et al., 2021), and fine-tuning subsets of the model (Zaken et al., 2021). One particularly promising approach is the use of low-rank adapters, which have demonstrated effectiveness in adapting models to new tasks with minimal additional parameters. Building upon LoRA, we propose an efficient continual learning neural architecture in this work. Our approach involves layering low-rank adapters on the key and value projection matrices of transformer blocks. By leveraging the benefits of low-rank adapters, we aim to strike a balance between model performance and computational efficiency in the context of continual learning.

## 6 Conclusion

In this paper, we introduce O-LoRA, a novel approach that leverages orthogonal subspace learning for continual learning in language models. O-LoRA systematically addresses catastrophic forgetting by adopting an incremental learning strategy within orthogonal subspaces. Distinguished by its data privacy considerations, efficient model parameter utilization, and robust generalization to novel tasks, our method stands out. Empirical evaluations underscore O-LoRA's efficacy in tackling the intricacies of continual learning.

## Limitations

While our method has demonstrated effectiveness in empirical evaluations, there are a few limitations to consider. Firstly, its performance and applicability in more complex scenarios with a large number of tasks, such as hundreds of tasks, require further investigation. Additionally, although our method does not rely on task identification during inference, it still requires task identification during training to train different LoRA parameters for each task. Exploring methods for task-agnostic training would be a valuable future direction. By addressing these limitations, we can enhance the scalability and task-agnostic capabilities of our approach, further advancing the field of continual learning for language models.

## Acknowledgements

The authors express their gratitude to the anonymous reviewers for their insightful comments. We also wish to acknowledge Hang Yan. Even though he wasn't directly involved in this study, his foundational guidance in continual learning has been pivotal to our research. This work was partially funded by Shanghai Academic Research Leader Program 22XD1401100.

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

# A Appendix

## A.1 Implementation Details

All of our experiments on t5 models were conducted on a machine equipped with 8 NVIDIA GeForce RTX 3090 and were implemented using DeepSpeed repository. For all orders of task streams, We trained the models with one epoch, a constant learning rate of 1e-3, a batch size of 64(a batch size of 8 per GPU), a dropout rate of 0.1, and a weight decay rate of 0. Only the values of $\lambda_1$ and $\lambda_2$ are different among order 1 to 6. For order 1, order 2 and order 3, we set $\lambda_1$ = 0.5, 0.5, 0.5, 0.5, $\lambda_2$ = 0, 0, 0, 0. For every task in order 4(MNLI, CB, WiC, COPA, QQP, BoolQA, RTE, IMDB, Yelp, Amazon, SST-2, DBpedia, Agnews, MultiRC, Yahoo), we set $\lambda_1$ = 0.5, 0.5, 0.5, 0.5, 0.5, 0.5, 0.5, 0.5, 0.5, 0.5, 0.5, 5, 5, 5, 5, and $\lambda_2$ = 0, 0, 0.1, 0, 0, 0, 0.3, 0.1, 0.05, 0, 0.1, 0.1, 0.1, 0, 0.1 respectively. For order 5(MultiRC, BoolQA, WiC, MNLI, CB, COPA, QQP, RTE, IMDB, SST-2, DBpedia, Agnews, Yelp, Amazon, Yahoo), we set $\lambda_1$ = 5, 5, 5, 5, 5, 5, 5, 5, 5, 5, 5, 5, 5, 5, 5, and $\lambda_2$ = 0, 0.1, 0, 0.1, 0.1, 0, 0.1, 0.3, 0.1, 0.5, 0, 0.1, 0, 0.1, 0.1 respectively. For order 6(Yelp, Amazon, MNLI, CB, COPA, QQP, RTE, IMDB, SST-2, DBpedia, Agnews, Yahoo, MultiRC, BoolQA, WiC), we set $\lambda_1$ = 0.5, 0.5, 0.02, 0.5, 0.5, 0.5, 0.5, 0.5, 0.5, 0.5, 0.5, 0.5, 0.5, 0.5, 0.5, and $\lambda_2$ = 0, 0, 0, 0.1, 0, 0, 0.3, 0, 0.1, 0.1, 0, 0.1, 0, 0.1, 0.3 respectively.

## A.2 Datasets

Table 4 shows details of the 15 datasets we used for our CL experiments, along with their evaluation metrics. Overall, we used datasets from CL benchmark (Zhang et al., 2015), GLUE (Wang et al., 2018) and SuperGLUE (Wang et al., 2019) benchmarks, and added IMDB movie reviews dataset, following (Razdaibiedina et al., 2023).

## A.3 Task Sequence Orders

We report task orders used for our CL experiments across T5 and LLaMA models in Table 5.

## A.4 Task Instructions

Table 6 shows prompts for different tasks. NLI denotes natural language inference, including MNLI, RTE and CB. SC denotes sentiment analysis, including Amazon, Yelp, SST-2 and IMDB. TC denotes topic classification, including AG News, Dbpedia and Yahoo.

## A.5 Detailed results of MMLU Zero-shot

| Dataset name | Category | Task | Domain | Metric |
|---|---|---|---|---|
| 1. Yelp | CL Benchmark | sentiment analysis | Yelp reviews | accuracy |
| 2. Amazon | CL Benchmark | sentiment analysis | Amazon reviews | accuracy |
| 3. DBpedia | CL Benchmark | topic classification | Wikipedia | accuracy |
| 4. Yahoo | CL Benchmark | topic classification | Yahoo Q&A | accuracy |
| 5. AG News | CL Benchmark | topic classification | news | accuracy |
| 6. MNLI | GLUE | NLI | various | accuracy |
| 7. QQP | GLUE | paragraph detection | Quora | accuracy |
| 8. RTE | GLUE | NLI | news, Wikipedia | accuracy |
| 9. SST-2 | GLUE | sentiment analysis | movie reviews | accuracy |
| 10. WiC | SuperGLUE | word sense disambiguation | lexical databases | accuracy |
| 11. CB | SuperGLUE | NLI | various | accuracy |
| 12. COPA | SuperGLUE | QA | blogs, encyclopedia | accuracy |
| 13. BoolQA | SuperGLUE | boolean QA | Wikipedia | accuracy |
| 14. MultiRC | SuperGLUE | QA | various | accuracy |
| 15. IMDB | SuperGLUE | sentiment analysis | movie reviews | accuracy |

Table 6: The details of 15 datasets used in our CL experiments. NLI denotes natural language inference, QA denotes questions and answers task. First five tasks correspond to the standard CL benchmark, all other tasks are used in long-sequence experiments.

| Order | Model | Task Sequence |
|---|---|---|
| 1 | T5, LLaMA | dbpedia → amazon → yahoo → ag |
| 2 | T5, LLaMA | dbpedia → amazon → ag → yahoo |
| 3 | T5, LLaMA | yahoo → amazon → ag → dbpedia |
| 4 | T5 | mnli → cb → wic → copa → qqp → boolqa → rte → imdb → yelp → amazon → sst-2 → dbpedia → ag → multirc → yahoo |
| 5 | T5 | multirc → boolqa → wic → mnli → cb → copa → qqp → rte → imdb → sst-2 → dbpedia → ag → yelp → amazon → yahoo |
| 6 | T5 | yelp → amazon → mnli → cb → copa → qqp → rte → imdb → sst-2 → dbpedia → ag → yahoo → multirc → boolqa → wic |

Table 7: Six different orders of task sequences used for continual learning experiments. Orders 1-3 correspond to the standard CL becnhmark adopted by prior works. Orders 4-6 are long-sequence orders spanning 15 tasks, following (Razdaibiedina et al., 2023).

| Task | Prompts |
|---|---|
| NLI | What is the logical relationship between the "sentence 1" and the "sentence 2"? Choose one from the option. |
| QQP | Whether the "first sentence" and the "second sentence" have the same meaning? Choose one from the option. |
| SC | What is the sentiment of the following paragraph? Choose one from the option. |
| TC | What is the topic of the following paragraph? Choose one from the option. |
| BoolQA | According to the following passage, is the question true or false? Choose one from the option. |
| MultiRC | According to the following passage and question, is the candidate answer true or false? Choose one from the option. |
| WiC | Given a word and two sentences, whether the word is used with the same sense in both sentence? Choose one from the option. |

Table 8: Instructions for different tasks.

| MMLU-task | LLaMA-7B | Alpaca-LoRA | Alpaca-LoRA-CL | Alpaca-inc-LoRA-CL | Alpaca-OLoRA-CL |
|---|---|---|---|---|---|
| math | 27 | 25.9 | 20.4 | 23.4 | 21.6 |
| health | 38.2 | 40.9 | 24.8 | 30.7 | 36.1 |
| physics | 30.8 | 32.5 | 22.2 | 27.7 | 30.3 |
| business | 40.3 | 50.3 | 25.2 | 32.3 | 38.7 |
| biology | 33.5 | 38.1 | 21.8 | 29.1 | 35.7 |
| chemistry | 25.4 | 27.7 | 17.2 | 20.1 | 23.8 |
| computer science | 30.6 | 35 | 27.2 | 30.8 | 31.3 |
| economics | 31.3 | 31.4 | 23 | 27 | 28 |
| engineering | 28.3 | 32.4 | 22.8 | 23.4 | 24.1 |
| philosophy | 32.6 | 34.5 | 23 | 26.1 | 32.1 |
| other | 37.2 | 46.9 | 24.6 | 33.5 | 42.1 |
| history | 39 | 45.8 | 25.1 | 33.3 | 42.3 |
| geography | 31.8 | 44.4 | 18.2 | 25.8 | 33.3 |
| politics | 37.7 | 40.1 | 21.8 | 28.2 | 33.6 |
| psychology | 39.4 | 40.4 | 22.8 | 28.3 | 35 |
| culture | 40.1 | 49.1 | 25.9 | 32.5 | 41 |
| law | 31.9 | 32.5 | 23.6 | 28.5 | 32.4 |
| Weighted Avg. | 34.4 | 37.5 | 23.3 | 28.6 | 33.6 |

Table 9: Detailed zero-shot results on MMLU benchmark of different CL methods.