# OpenReview forum: "Orthogonal Subspace Learning for Language Model Continual Learning"
_EMNLP/2023/Conference — EMNLP 2023 Findings_

### Official Review · Reviewer_7Yor · 2023-07-27

**Soundness:** 4

**Excitement:**

4: Strong: This paper deepens the understanding of some phenomenon or lowers the barriers to an existing research direction.

**Paper Topic And Main Contributions:**

This paper explores combining instructions tuning with orthogonal LoRA for multi-tasks learning. The authors keep each LoRA vector subspaces learnt for each task orthogonal to each others, with the aim of limiting forgetting previous tasks when learning a new one.

**Questions For The Authors:**

Please refer to reasons to reject above.

**Reasons To Accept:**

The field of learning new tasks without leading to catastrophic forgetting is an interesting and challenging one. The proposed method seems to yield strong results.

**Reasons To Reject:**

The main idea behind this paper seems to be flawed. Indeed, the authors argue that learning new tasks can lead to catastrophic forgetting, and that using orthogonal LoRA is necessary to avoid it. However, in the present setting, each task is learnt using its own set of LoRA matrices. How can the model forget, then, when its parameters are kept fixed? Why do we need each LoRA subspaces to be orthogonal, as, when we learn a new task, we keep every parameters linked to other tasks fixed?

Another major issue comes from the fact that this method is actually a combination of two: instructions tuning and orthogonal LoRA. Yet no ablation study seems to have been performed to investigate the contribution of each of them. My intuition is that orthogonal LoRA should perform similarly as regular LoRA on its own, given the idea described above.

Edit: the authors have clarified the above objections, I have updated my scores accordingly.

**Reproducibility:**

4: Could mostly reproduce the results, but there may be some variation because of sample variance or minor variations in their interpretation of the protocol or method.

**Reviewer Confidence:**

4: Quite sure. I tried to check the important points carefully. It's unlikely, though conceivable, that I missed something that should affect my ratings.

---

> ### Author Rebuttal · Authors · 2023-08-28
>
> Thank you for taking the time to review our paper and for providing feedback.
>
> We kindly request a few minutes of your time to carefully consider our clarifications and responses.
>
>
>
> > R1: **About  OLoRA VS separate LoRA**.
>
> Let's address the core concern first, specifically highlighting the advantages of our continual learning method over training separate LoRA models for different tasks.
>
> **1. Real-World Dynamic Environments:** In real-world scenarios, new tasks or variations of tasks often emerge over time. Training a separate LoRA model every time is both computationally costly and impractical, especially when quick adaptation is essential.
>
> **2. Knowledge Transfer:** Continual learning allows a model to transfer knowledge from one task to another. This can often lead to improved performance on new tasks, especially when the new tasks are related to previously learned tasks. Separate models don't have this advantage.
>
> **3. User Interface:** O-LoRA integrates all tasks into one model, interacting only through instructions, bypassing the need for task IDs. In contrast, task-specific LoRA demands explicit task IDs during inference, which restricts its usability and practicality.
>
> **4. Inference Efficiency :** Using separate models for each task, even with LoRA, can significantly increase inference time. Consider a system like ChatGPT: if every query required a different model, the response time would be impractical. Continual learning consolidates this, ensuring faster and efficient answers without switching between models.
>
> In conclusion, while it is technically possible to train separate models for each task, Continual learning  [1] [2] presents a more efficient, practical, and sustainable approach, especially for scenarios with evolving tasks and constraints.
>
>
>
> **About catastrophic forgetting**
>
> Catastrophic forgetting, as described in [3] [4], is a phenomenon wherein a neural network, while learning new tasks, inadvertently overwrites or conflicts with the knowledge from earlier tasks. This interference results in reduced performance on those initial tasks. It is imperative to address this issue [1] [2], as neglecting it undermines the core principle of continual learning: the capacity to consistently amass knowledge without detrimentally affecting prior knowledge. Research by [6] illustrates that as the skills of models like ChatGPT expand, their proficiency in previously acquired skills tends to degrade.
>
>
>
> > R2: About ablation study of instructions tuning and orthogonal LoRA.
>
> **We didn't exclude the instruction in our experiments due to its role as a soft task identifier.** Given the same input, the intended tasks (e.g., sentiment classification, entity recognition, translation) could vary. Without the instruction, the language model would lack task specificity. Hence, it's essential for our approach.**To ensure fairness and account for potential instruction effects, all baseline methods utilized the same instruction format as O-LoRA.** The results presented in Table 2 demonstrate that the observed enhancement from O-LoRA is attributable to the method itself, rather than the influence of the instruction.
>
> In Table 2, we present a comparison between two baselines: the method of individually training LoRA for each task (denoted as PerTaskFT) and the method of sequential LoRA learning (denoted as SeqLoRA). Let's revisit these results for clarity:
>
> ```markdown
> |                    |             Standard CL Benchmark         |
> |                    | Order-1      | Order-2     | Order-3      |
> | ------------------ | ------------ | ----------- | ------------ |
> | SeqLoRA            | 44.6         | 32.7        | 53.7         |
> | PerTaskFT          | 70.0         | 70.0        | 70.0         |
> | O-LoRA             | 77.0         | 77.2        | 77.1         |
> ```
>
> **SeqLoRA vs. O-LoRA**: The unconstrained sequential task learning approach, SeqLoRA, consistently underperforms compared to O-LoRA. This stark contrast underscores the severity of the catastrophic forgetting phenomenon, emphasizing its importance and urging attention.
>
> **PerTaskFT vs. O-LoRA**: When contrasted with the PerTaskFT method, which trains LoRA separately for each task, O-LoRA capitalizes on inter-task knowledge correlations. This leads to enhanced transfer capabilities and consequently better overall performance.
>
>
>
>
>
> **Key References :**
>
> - [1] Wang, Liyuan, et al. "A comprehensive survey of continual learning: Theory, method and application." arXiv preprint arXiv:2302.00487 (2023).
> - [2] Ke, Zixuan, and Bing Liu. "Continual learning of natural language processing tasks: A survey." *arXiv preprint arXiv:2211.12701* (2022).
> - [3] McClelland, James L., Bruce L. McNaughton, and Randall C. O'Reilly. "Why there are complementary learning systems in the hippocampus and neocortex: insights from the successes and failures of connectionist models of learning and memory." *Psychological review* 102.3 (1995): 419.
> - [4] McCloskey, Michael, and Neal J. Cohen. "Catastrophic interference in connectionist networks: The sequential learning problem." *Psychology of learning and motivation*. Vol. 24. Academic Press, 1989. 109-165.
> - [5] Wu, Tongtong, et al. "Pretrained language model in continual learning: A comparative study." *International Conference on Learning Representations*. 2021.
> - [6] Chen, Lingjiao, Matei Zaharia, and James Zou. "How is ChatGPT's behavior changing over time?." *arXiv preprint arXiv:2307.09009* (2023).

---

### Official Review · Reviewer_GrPc · 2023-08-05

**Soundness:** 2

**Excitement:**

3: Ambivalent: It has merits (e.g., it reports state-of-the-art results, the idea is nice), but there are key weaknesses (e.g., it describes incremental work), and it can significantly benefit from another round of revision. However, I won't object to accepting it if my co-reviewers champion it.

**Paper Topic And Main Contributions:**

This paper addresses the problem of catastrophic fogetting in continual learning. The authors propose O-LoRA, which trains a task-specific low-rank adapter in an orthogonal direction from the previously learned adapters to minimize the effect to them. The experimental results in the paper show that the proposed method outperforms various continual learning methods.

**Reasons To Accept:**

A. The paper is well-written and easy to follow.

B. The soundness of the proposed method is sufficiently backed up by previous work.

C. The experimental results present several interesting results, e.g., continually learned O-LoRA outperforms task-specific LoRA, which has not been observed in the standard fine-tuning scenarios.

**Reasons To Reject:**

A. The comparison with the state-of-the-art continual learning method [1] is missing. Particularly, in some evaluations, the state-of-the-art method outperforms the proposed method.

B. The paper lacks in-depth analysis and comparison, including hyper-parameter optimality / sensitivity, forward / backward transfer, and generalization effects to unseen sequential tasks (as the authors mentioned that these effects were a unique strength of the proposed method).

[1] Razdaibiedina et al., Progressive Prompts: Continual Learning for Language Models, ICLR 2023.

**Reproducibility:**

3: Could reproduce the results with some difficulty. The settings of parameters are underspecified or subjectively determined; the training/evaluation data are not widely available.

**Reviewer Confidence:**

4: Quite sure. I tried to check the important points carefully. It's unlikely, though conceivable, that I missed something that should affect my ratings.

---

> ### Author Rebuttal · Authors · 2023-08-28
>
> Thank you for your constructive feedback on our paper. We truly appreciate the time you invested in the review. We've carefully considered your insights and addressed the highlighted concerns. We hope our responses shed more light on the matter.
>
>
>
> > RA: The comparison with the state-of-the-art continual learning method [1] is missing. Particularly, in some evaluations, the state-of-the-art method outperforms the proposed method.
>
> We appreciate your observation. In our initial submission, we overlooked the comparison with [1] due to its approach of training distinct models for each task. While such models excel in tasks they are trained for, their performance on unseen tasks can be inconsistent.
>
> However, recognizing the importance of a comprehensive evaluation, we will incorporate comparisons with [1] in future revisions.
>
>
>
> > RB:  The paper lacks in-depth analysis and comparison, including hyper-parameter optimality / sensitivity, forward / backward transfer, and generalization effects to unseen sequential tasks (as the authors mentioned that these effects were a unique strength of the proposed method).
>
> Thank you for raising these concerns.
>
> Regarding **generalization to unseen tasks**, section 4.2 of our paper delves deeply into this. Our findings are illustrated in **Table 3**:
>
> ```markdown
> |                    | MMLU         | CL Benchmark|
> | ------------------ | ------------ | ----------- |
> | LLaMA-7B           | 34.4         | /           |
> | Alpaca-LoRA        | 37.5         | /           |
> | Alpaca-LoRA-CL     | 23.3         | 46.7        |
> | Alpaca-inc-LoRA-CL | 28.6         | 33.1        |
> | ------------------ | ------------ | ----------- |
> | Alpaca-OLoRA-CL    | 78.9         | 79.0        |
> ```
>
> We utilized the MMLU benchmark, which is tailored for assessing generalization to such tasks.  As evidenced by our results, our method not only outperforms the baseline on new tasks but also maintains the generalization capabilities of the LLM. For a detailed discussion, please refer to the section 4.2 segment titled "Impact on the Generalization Ability of LLMs."
>
> Appreciating your insightful feedback, we will expand our analysis in subsequent versions, adding insights on hyper-parameter optimality/sensitivity and introducing metrics for forward/backward transfer.

---

### Official Review · Reviewer_jxjZ · 2023-08-11

**Typos Grammar Style And Presentation Improvements:** 1. [Line 482] Optimal rank for O-LoRA…
**Soundness:** 2

**Excitement:**

2: Mediocre: This paper makes marginal contributions (vs non-contemporaneous work), so I would rather not see it in the conference.

**Missing References:**

[1] Mehta, Sanket Vaibhav, Darshan Patil, Sarath Chandar, and Emma Strubell. "An Empirical Investigation of the Role of Pre-training in Lifelong Learning." (2021).

[2] Razdaibiedina, Anastasia, Yuning Mao, Rui Hou, Madian Khabsa, Mike Lewis, and Amjad Almahairi. "Progressive Prompts: Continual Learning for Language Models." In The Eleventh International Conference on Learning Representations. 2022.

[3] Wang, Zhicheng, Yufang Liu, Tao Ji, Xiaoling Wang, Yuanbin Wu, Congcong Jiang, Ye Chao et al. "Rehearsal-free Continual Language Learning via Efficient Parameter Isolation." In Proceedings of the 61st Annual Meeting of the Association for Computational Linguistics (Volume 1: Long Papers), pp. 10933-10946. 2023.

[4] Yu, Tianhe, Saurabh Kumar, Abhishek Gupta, Sergey Levine, Karol Hausman, and Chelsea Finn. "Gradient surgery for multi-task learning." Advances in Neural Information Processing Systems 33 (2020): 5824-5836.


**Paper Topic And Main Contributions:**

The paper tackles the challenge of continual learning in the context of pre-trained models, where tasks are encountered sequentially. Specifically, it focuses on the phenomenon of catastrophic forgetting, which arises when the continual learning of new tasks disrupts the performance of previously acquired tasks, leading to a decline in their effectiveness. The paper approaches the problem by employing the widely used parameter-efficient strategy, such as LORA, to facilitate the learning of new tasks and the overall technique involves training distinct sets of LORA parameters for each task. To address the problem of interference between the new and previously learned LORA parameters, the paper introduces an extension to the Orthogonal Gradient Descent method. This extension involves learning new tasks within distinct low-rank vector subspaces that remain orthogonal to one another, thereby minimizing interference. The effectiveness of this approach is demonstrated through empirical results on two distinct benchmarks, showcasing its superiority over the existing baseline.

**Questions For The Authors:**

A. What are the outcomes when evaluating O-LoRA's performance without the inclusion of the instruction tuning scheme? How do the baseline methods fare when augmented with the instruction tuning scheme? To what extent does the instruction tuning paradigm contribute to mitigating forgetting?

B. How does LoRA's effectiveness manifest when integrated with the considered baselines such as EWC, Replay, and LwF? In comparison to rehearsal-free methodologies like SAM-based optimization [1] and dynamically expandable architectures utilizing progressive prompts [2], and parameter isolation [3] how does the proposed approach perform?

C. The paper only reports average performance after training on the last task in the sequences. What is the behavior of the performance of the earlier tasks over the course of continual learning? Additionally, quantifying the forgetting metrics alongside the average performance, as demonstrated in [1] for forgetting and learning accuracy metrics, would provide insightful context.

D. In the caption of Table 1, it is stated that TIF denotes the availability of task-id during inference. The rationale behind why EWC, A-GEM, MbPA++, etc., require task-id during inference remains unclear. Is this an oversight or are there specific details that have not been provided?

E. In Figure 4 of the analysis, the paper visualizes the variation in hidden states across different layers with/ without O-LoRA constraints. Please specify how is this variation computed? Also, in case of the encoder, there is increased variation for both the considered scenarios. How does this align with the forgetting differences? Also, what makes the paper believe that less variation corresponds to lesser forgetting? Is there any evidence about smoothness of the representation?

F. The paper contrasts the performance of various Pre-trained Language Models (PLMs) and draws the conclusion that larger networks mitigate catastrophic forgetting. However, there appears to be a confounding factor – the diversity of pre-training corpora across different PLMs. This outcome is already widely acknowledged within the community, as indicated in [1].


**Reasons To Accept:**

1. The paper addresses an important problem of sequential task learning, particularly within pre-trained models. It presents a realistic continual learning scenario involving privacy and storage constraints, and unavailability of the task identity during the inference.

2. Toward the above-mentioned setup, the paper proposes an efficient solution based upon the widely adopted parameter-efficient finetuning technique like LORA. Also, the proposed Orthogonal LORA is efficient compared to previous rehearsal-based methods as it does not require increased computation as the number of tasks increases in the sequences, except for additional complexity from the auxiliary loss computation to enforce orthogonality constraints.

3. The paper is well-written and easy to follow.


**Reasons To Reject:**

1. One of the major weaknesses lies in the inaccurate assertion that the proposed O-LoRA can be utilized to address unseen tasks more effectively than established baselines such as EWC, A-GEM, and others. The paper adopts widely used instruction tuning as the training paradigm for O-LoRA which enables generalization to unseen/ novel tasks. To ensure a fair assessment against the considered baselines, it is imperative for the paper to apply instruction tuning uniformly across all methods. The inconsistency in the training methodology between O-LoRA and the other baselines skews the outcomes presented in Table 2, where O-LoRA exhibits notably superior performance. To enhance the rigor of experimentation, a concrete suggestion is as follows: initially compare O-LoRA without the instruction tuning paradigm against the baselines. Subsequently, introduce instruction tuning to all methods and observe the ensuing improvements. This approach would not only clarify the role played by the instruction tuning paradigm in addressing catastrophic forgetting but also yield insights into the efficacy of different continual learning methods.
Another claim the paper makes is that the proposed approach is parameter efficient as it uses the LoRA technique for sequential fine-tuning.

2. By definition, the proposed solution experiences less forgetting as only a minimal set of parameters is updated while sequential learning. Conversely, methodologies like EWC, A-GEM, and LwF update all model parameters, thereby experiencing severe interference and increased forgetting. It is noteworthy that many of these methodologies originate from the period before the prevalence of pre-training and fine-tuning. In this context, the assertion that O-LoRA is parameter-efficient when measured against the baselines becomes untenable. Aligning with the earlier point, the paper should evaluate these methodologies by solely fine-tuning the LoRA parameters (ensuring comparability of trainable parameters across methods is vital for fair comparisons).

3. The paper fails to compare their method with other rehearsal-free methods like SAM-based optimization for continual learning [1], and dynamically expandable architectures for continual learning [2, 3].

4. The paper lacks specific information regarding implementation, apart from indicating the models employed for experimentation. It omits crucial details about dataset size, training specifics (such as batch size, learning rate, epoch count, learning rate decay, etc.), and hyperparameters associated with the considered baselines (like regularization strength/ lambda for EWC, memory buffer size for replay, etc.). This omission raises a concern regarding the ability to replicate the reported results.


**Reproducibility:**

2: Would be hard pressed to reproduce the results. The contribution depends on data that are simply not available outside the author's institution or consortium; not enough details are provided.

**Reviewer Confidence:**

5: Positive that my evaluation is correct. I read the paper very carefully and I am very familiar with related work.

---

> ### Author Rebuttal · Authors · 2023-08-28
>
> We appreciate your feedback and have carefully considered your suggestions. We try to address your remarks below:
>
>
>
> > **R1:** About inconsistency in comparing O-LoRA with established baselines by not uniformly applying instruction tuning across all methods.
>
> We'd like to clarify that during our experiments, **the baseline methods were trained using the same prompts as O-LoRA, ensuring uniformity in comparisons.**
>
> We recognize that our experimental section lacked explicit mention of this, leading to potential misunderstandings. In subsequent versions of the paper, we will incorporate detailed descriptions regarding the implementation of the baselines to provide clarity.
>
>
>
> > **R2:**  About directly comparing O-LoRA's efficiency against these baselines without ensuring comparability of trainable parameters is not a fair comparison.
>
> It's essential to note that **among the baselines compared in the paper, except for prompt-based methods such as L2P and LFPT5, all others are implemented based on LoRA**. This ensures a consistent parameter setting across O-LoRA and these baselines, facilitating an fair comparison.
>
> We acknowledge the oversight in not detailing this in the original paper. We will furnish details regarding this aspect of baseline implementation in the upcoming revision of the paper. Thank you for highlighting this concern.
>
>
>
> > **R3 & QB:** About compare with SAM-based optimization and dynamically expandable architectures methods.
>
> In our initial submission, we omitted comparisons with dynamically expandable architectures, primarily because these methods tend to train specific models for each task. Although they often excel on tasks they've been trained on, their effectiveness on unseen tasks is questionable.
>
> However, recognizing the importance of a comprehensive evaluation, we will incorporate comparisons with SAM-based optimization and dynamically expandable architectures in future revisions.
>
>
>
> > **R4:** About missing details about dataset size, training specifics (such as batch size, learning rate, epoch count, learning rate decay, etc.)
>
> We have referenced the benchmark datasets at lines 304, 313, and 330 in our paper, pointing to the original publications where they were introduced.
>
> We recognize the oversight regarding the detailed training specifics. In our subsequent revision, we will explicitly mention the training hyperparameters: a batch size of 64, learning rate set at 1e-03, epoch count at 1, and a weight decay of 0. Furthermore, we will make the code and related datasets publicly available concurrently with the paper's open access, ensuring reproducibility of our experimental results.
>
>
>
> > **QA**: What are the outcomes when evaluating O-LoRA's performance without the inclusion of the instruction tuning scheme? How do the baseline methods fare when augmented with the instruction tuning scheme? To what extent does the instruction tuning paradigm contribute to mitigating forgetting?
>
> We didn't exclude the instruction in our experiments due to its role as a soft task identifier. **Given the same input, the intended tasks (e.g., sentiment classification, entity recognition, translation) could vary. Without the instruction, the language model would lack task specificity.** Hence, it's essential for our approach.
>
> To control for instruction effects, all baseline methods employed the same instruction as O-LoRA. The results presented in Table 2 demonstrate that the observed enhancement from O-LoRA is attributable to the method itself, rather than the influence of the instruction.
>
>
>
> > **QC**: The paper only reports average performance after training on the last task in the sequences. What is the behavior of the performance of the earlier tasks over the course of continual learning? Additionally, quantifying the forgetting metrics alongside the average performance, as demonstrated in [1] for forgetting and learning accuracy metrics, would provide insightful context.
>
> Thank you for the insightful suggestion. In subsequent revisions, we plan to incorporate additional metrics, including forward transfer score [2], backward transfer score [2], and forgetting metrics [1], to offer a more comprehensive perspective on the model's behavior during continual learning.
>
>
>
> > **QD**: In the caption of Table 1, it is stated that TIF denotes the availability of task-id during inference. The rationale behind why EWC, A-GEM, MbPA++, etc., require task-id during inference remains unclear. Is this an oversight or are there specific details that have not been provided?
>
> EWC, A-GEM, and MbPA++ were originally implemented based on classification models like Bert. Consequently, during the inference phase, they require a task-id to utilize the corresponding classification head for generating predictions for a given task.
>
>
>
> > **QE**: In Figure 4 of the analysis, the paper visualizes the variation in hidden states across different layers with/ without O-LoRA constraints. Please specify how is this variation computed? Also, in case of the encoder, there is increased variation for both the considered scenarios. How does this align with the forgetting differences? Also, what makes the paper believe that less variation corresponds to lesser forgetting? Is there any evidence about smoothness of the representation?
>
> Figure 4 illustrates **how the hidden states of previously tasks change after introducing a new task to the model.** Specifically, for a sequence like dbpedia (trained task) -> amazon (new task), the model trained on dbpedia is represented as $\theta_d$ and post training on amazon becomes $\theta_{da}$. The variation is quantified by evaluating the change in the representations for the dbpedia validation set $x_d$, i.e., $\theta_{da}(x_d)-\theta_d(x_d)$. A larger variation implies that the model extracts different features.
>
> For the encoder, both approaches demonstrate significant representation changes, suggesting that both are extracting new features. However, this doesn't necessarily confirm if the model has retained the pertinent features from previous tasks.
>
> On the decoder side, it determines which features the model should rely on for predictions. Lesser variation in decoder representations indicates that for previous tasks, the important features encoded by the model are largely preserved, which implies reduced forgetting.
>
> We plan to introduce an analysis similar to that in [1] in our subsequent version to offer a deeper insight into this aspect.
>
>
>
> > **QF**: The paper contrasts the performance of various Pre-trained Language Models (PLMs) and draws the conclusion that larger networks mitigate catastrophic forgetting. However, there appears to be a confounding factor – the diversity of pre-training corpora across different PLMs.
>
> We appreciate the reviewer's observation regarding the potential confounding factor of the diversity of pre-training corpora across different PLMs.
>
> However, **our conclusion is drawn based on a comparison of T5 models of varying sizes, all of which were trained on the same corpus**. Thus, this concern is not applicable in our case. As highlighted in Table 4, we re-emphasize our experimental results to clarify this point.
>
> ```markdown
> |                    |             Standard CL Benchmark         |
> |                    | Order-1      | Order-2     | Order-3      |
> | ------------------ | ------------ | ----------- | ------------ |
> | T5-base            | 73.9         | 75.8        | 74.5         |
> | T5-large           | 77.0         | 77.2        | 77.1         |
> | T5-xl              | 78.9         | 79.0        | 77.19        |
> ```
>
>
>
> > **Missing References:**
>
> We appreciate your feedback. While our paper already references [2] and [3], we will incorporate [1] and [4] in our subsequent version and conduct a detailed comparison with them.
>
>
>
> > **Typos Grammar Style**
>
> Thank you for pointing out these issues. We will make the necessary corrections in our next version.

---

### Official Review · Reviewer_PwhU · 2023-08-13

**Soundness:** 4

**Excitement:**

4: Strong: This paper deepens the understanding of some phenomenon or lowers the barriers to an existing research direction.

**Missing References:**

To my knowledge, the literature review is fine, but it would be even better to include more references in class-incremental language learning, e.g., continual relation extraction and continual intent detection.

[1] Han, Xu, et al. "Continual relation learning via episodic memory activation and reconsolidation." Proceedings of the 58th Annual Meeting of the Association for Computational Linguistics. 2020.
[2] Wu, Tongtong, et al. "Curriculum-meta learning for order-robust continual relation extraction." Proceedings of the AAAI Conference on Artificial Intelligence. Vol. 35. No. 12. 2021.
[3] Bai, Guirong, et al. "Incremental intent detection for medical domain with contrast replay networks." Findings of the Association for Computational Linguistics: ACL 2022. 2022.
[4] Chen, Xiudi, Hui Wu, and Xiaodong Shi. "Consistent Prototype Learning for Few-Shot Continual Relation Extraction." Proceedings of the 61st Annual Meeting of the Association for Computational Linguistics (Volume 1: Long Papers). 2023.

**Paper Topic And Main Contributions:**

The paper titled "Orthogonal Low-Rank Adaptation for Continual Learning in Language Models" introduces a novel approach, O-LoRA, aimed at addressing the issue of catastrophic forgetting in pre-trained language models when encountering multiple tasks sequentially. The authors propose a solution that leverages orthogonal low-rank adaptation to mitigate the performance degradation that typically occurs in such scenarios.

**Questions For The Authors:**

The author suggests a continuous addition of new Lora components upon the emergence of new tasks, which introduces interesting aspects and prompts the following inquiries:

Can the proposed Orthogonal Subspace Learning technique effectively enhance the learning process for a sequence of similar tasks? Although the method exhibits strong performance in mitigating forgetting issues, does it also facilitate the transfer of knowledge between these similar tasks?

With the linear increase in parameters as new tasks are incorporated, does the timing of task learning influence performance, potentially resulting in poorer outcomes for tasks learned later? Is there a discernible relationship between the order of tasks and the proportion of learnable parameters within the overall parameter count?



**Reasons To Accept:**

- Effective mitigation of Catastrophic Forgetting
- Well-structured and easy to follow
- Minimal parameter costs and no user data storage

**Reasons To Reject:**

- Code is unavailable

**Reproducibility:**

3: Could reproduce the results with some difficulty. The settings of parameters are underspecified or subjectively determined; the training/evaluation data are not widely available.

**Reviewer Confidence:**

5: Positive that my evaluation is correct. I read the paper very carefully and I am very familiar with related work.

**Typos Grammar Style And Presentation Improvements:**

Minor Issue:
In line 420, the acronym "LLM" appears to refer to the Large Language Model, rather than the intended reference to the Language Learning Model.

---

> ### Author Rebuttal · Authors · 2023-08-27
>
> Thank you for your appreciation and an excellent summarization of our work.  We appreciate your time and effort in carefully reviewing our work. We try to address your remarks below:
>
>
>
> > **R1**: Code is unavailable
>
> We acknowledge the concern. We commit to releasing the code and associated data simultaneously when the paper is made public.
>
>
>
> > **Q1:** Can the proposed Orthogonal Subspace Learning technique effectively enhance the learning process for a sequence of similar tasks? Although the method exhibits strong performance in mitigating forgetting issues, does it also facilitate the transfer of knowledge between these similar tasks?
>
> Certainly, **O-LoRA possesses the capability to transfer knowledge between different tasks**. This assertion is substantiated by our empirical results presented in Table 2, where the performance of O-LoRA surpasses that of "PerTaskFT". The latter method involves training a model specifically for each task and then computing their average accuracy. Below, we re-highlight the experimental results:
>
> ```markdown
> |                    |             Standard CL Benchmark         |
> |                    | Order-1      | Order-2     | Order-3      |
> | ------------------ | ------------ | ----------- | ------------ |
> | PerTaskFT          | 70.0         | 70.0        | 70.0         |
> | O-LoRA             | 77.0         | 77.2        | 77.1         |
> ```
>
> Nevertheless, on the benchmark involving a large number of tasks, the performance of PerTaskFT excels. This discrepancy can be attributed to the fact that the benefits of knowledge transfer are outweighed by the catastrophic forgetting associated with longer task sequences. In the updated version of our paper, we plan to incorporate Forward Transfer Scores to provide further details on this aspect.
>
>
>
> > **Q2:** With the linear increase in parameters as new tasks are incorporated, does the timing of task learning influence performance, potentially resulting in poorer outcomes for tasks learned later? Is there a discernible relationship between the order of tasks and the proportion of learnable parameters within the overall parameter count?
>
> An insightful query. Indeed, the sequential placement of a task exerts an influence on its learning efficacy. When a task is positioned earlier in a sequence, it generally manifests more optimal fitting results than if positioned later. Yet, this performance decline becomes conspicuously accentuated in particularly protracted task sequences.  Addressing continual learning in such extended sequences remains an open challenge.
>
> To mitigate the suboptimal outcomes experienced by tasks introduced later in the sequence, a pragmatic approach would involve attenuating the stringent orthogonality constraints applied to them.
>
>
>
> > **Missing References:**
>
> Thank you for the suggestion. The highlighted works, focusing on continual learning in specific NLP tasks, indeed represent significant contributions and provide valuable supplements to general continual learning methodologies. We will add these references in subsequent versions of our paper.
>
>
>
> > **Typos Grammar Style**
>
> Thanks, we will correct this error in the revised paper.

---

### Meta-Review · Area_Chair_Frz2 · 2023-09-13

**Recommendation:** 3

**Metareview:**

The authors propose O-LoRA, a new method for continual learning with large language models. For each new task, a new set of LoRA parameters is learned, but, different than the straightforward IncLoRA baseline, an orthogonality regularizer encourages the LoRA parameters to add new information compared to prior sets of parameters. Experiments vs. baselines demonstrate promising performance improvements. In addition, the proposed method is parameter efficient, and doesn't require task replay, which are important practical considerations, e.g., for privacy reasons.

Reviewers were mixed in their assessments, even after significant discussion.

A summary of the more negative leaning reviews/associated discussion:

===
Reviewer jxjZ maintains some concerns, including: 1) are the baselines LoRA or not? --- the authors say that "except for prompt-based methods such as L2P and LFPT5, all others are implemented based on LoRA" but the reviewer reasonably points out that the most plausible read of the results "suggests that all model parameters are fine-tuned for EWC and Replay." (which might cause more forgetting than just using LoRA); 2) How will the promised SAM-based results compare?; and 3) Were hyperparameters like LoRA r were optimized for the baseline?
~
Reviewer GrPc shares jxjZ's concerns about hyperparameter configuration optimization, but also, mentions yet another missing method from ICLR 2023 from Razdaibiedina et al. 2023 (which the authors promise to compare to, but experiments are not finished).

===

Among these concerns, I find the points about the baselines most compelling: it's not clear if they are LoRA versions (the authors didn't reply to the reply from jxjZ) and, as both reviewers mention, it's not clear how hyperparameters of baselines were optimized, nor is it clear how the author's method will compare to several missing baselines which the reviewers expected.

Overall: the authors could likely address many of the above concerns in revision for a potential camera ready, but I am hesitant to give a higher rating without seeing 1) the additional clarifications requested by jxjZ; 2) more details of how the baselines were hyperparameter tuned; and 3) how the missing baselines perform.

---

### Decision · Program_Chairs · 2023-10-07

**Decision:**

Accept-Findings

**Comment:**

The authors propose O-LoRA, a new method for continual learning with large language models. For each new task, a new set of LoRA parameters is learned, but, different than the straightforward IncLoRA baseline, an orthogonality regularizer encourages the LoRA parameters to add new information compared to prior sets of parameters. Experiments vs. baselines demonstrate promising performance improvements. In addition, the proposed method is parameter efficient, and doesn't require task replay, which are important practical considerations, e.g., for privacy reasons.

Reviewers were mixed in their assessments, even after significant discussion.

A summary of the more negative leaning reviews/associated discussion:

===
Reviewer jxjZ maintains some concerns, including: 1) are the baselines LoRA or not? --- the authors say that "except for prompt-based methods such as L2P and LFPT5, all others are implemented based on LoRA" but the reviewer reasonably points out that the most plausible read of the results "suggests that all model parameters are fine-tuned for EWC and Replay." (which might cause more forgetting than just using LoRA); 2) How will the promised SAM-based results compare?; and 3) Were hyperparameters like LoRA r were optimized for the baseline?
~
Reviewer GrPc shares jxjZ's concerns about hyperparameter configuration optimization, but also, mentions yet another missing method from ICLR 2023 from Razdaibiedina et al. 2023 (which the authors promise to compare to, but experiments are not finished).

===

Among these concerns, I find the points about the baselines most compelling: it's not clear if they are LoRA versions (the authors didn't reply to the reply from jxjZ) and, as both reviewers mention, it's not clear how hyperparameters of baselines were optimized, nor is it clear how the author's method will compare to several missing baselines which the reviewers expected.

Overall: the authors could likely address many of the above concerns in revision for a potential camera ready, but I am hesitant to give a higher rating without seeing 1) the additional clarifications requested by jxjZ; 2) more details of how the baselines were hyperparameter tuned; and 3) how the missing baselines perform.